# Integration of In Silico Strategies for Drug Repositioning towards P38α Mitogen-Activated Protein Kinase (MAPK) at the Allosteric Site

**DOI:** 10.3390/pharmaceutics14071461

**Published:** 2022-07-13

**Authors:** Utid Suriya, Panupong Mahalapbutr, Thanyada Rungrotmongkol

**Affiliations:** 1Program in Biotechnology, Faculty of Science, Chulalongkorn University, Bangkok 10330, Thailand; noteutidii@gmail.com; 2Department of Biochemistry, Center for Translational Medicine, Faculty of Medicine, Khon Kaen University, Khan Kaen 40002, Thailand; panupma@kku.ac.th; 3Center of Excellence in Structural and Computational Biology, Department of Biochemistry, Chulalongkorn University, Bangkok 10330, Thailand; 4Ph.D. Program in Bioinformatics and Computational Biology, Graduate School, Chulalongkorn University, Bangkok 10330, Thailand

**Keywords:** drug repositioning, p38α MAPK, molecular docking, MD simulation, allosteric inhibitors, in silico screening, computer-aided drug discovery

## Abstract

P38α mitogen-activated protein kinase (p38α MAPK), one of the p38 MAPK isoforms participating in a signaling cascade, has been identified for its pivotal role in the regulation of physiological processes such as cell proliferation, differentiation, survival, and death. Herein, by shedding light on docking- and 100-ns dynamic-based screening from 3210 FDA-approved drugs, we found that lomitapide (a lipid-lowering agent) and nilotinib (a Bcr-Abl fusion protein inhibitor) could alternatively inhibit phosphorylation of p38α MAPK at the allosteric site. All-atom molecular dynamics simulations and free energy calculations including end-point and QM-based ONIOM methods revealed that the binding affinity of the two screened drugs exhibited a comparable level as the known p38α MAPK inhibitor (BIRB796), suggesting the high potential of being a novel p38α MAPK inhibitor. In addition, noncovalent contacts and the number of hydrogen bonds were found to be corresponding with the great binding recognition. Key influential amino acids were mostly hydrophobic residues, while the two charged residues including E71 and D168 were considered crucial ones due to their ability to form very strong H-bonds with the focused drugs. Altogether, our contributions obtained here could be theoretical guidance for further conducting experimental-based preclinical studies necessary for developing therapeutic agents targeting p38α MAPK.

## 1. Introduction

Mitogen-activated protein kinase (MAPK) signaling pathways are a cascade comprising three kinases including extracellular signal-regulated kinase (ERK), c-Jun NH_2_-terminal kinase (JNK), and p38, in which the upstream kinase (MAPKKK) responds to various extra- and intracellular signals and activates the middle kinase (MAPKK) by direct phosphorylation [1]. Then, MAPKKs phosphorylate and activate a MAPK, resulting in cell-specific physiological phenomena such as cell proliferation, differentiation, survival, and death [2]. MAPKs are known to be able to react with a wide range of input signals including hormones, cytokines, and growth factors, as well as endogenous stress and environmental factors. To this end, they were classified into two distinct responsive MAPKs; mitogen activated (ERK) and stress activated kinases (JNK and p38) [3]. Substantial studies revealed that the p38 pathway is a key player in response to environmental stress signals and inflammatory stimuli as well as being responsible for the production of some inflammatory cytokines such as tumor necrosis factor-α (TNF-α), interleukin-1β, interleukin-6, and interleukin-12 in response to proinflammatory signaling [4,5]. Furthermore, p38 can be a restraint in cancer tumorigenesis (e.g., breast, lung, colon, and liver cancer), which induces a p38-mediated proapoptotic mechanism and the killing of incipient tumor cells by a mechanism involved in the production of reactive oxygen species (ROS) [6]. However, p38 activity functions conversely once a tumor has already been established by supporting its growth [7]. Experimental evidence indicates that tumor cells need to modulate the level of p38 MAPK activity in order to perform metastases, and this signaling occurs in a variety of diseases [8]. To this end, inhibition of the p38 pathway has attracted much attention for the reason that it could be a promising strategy in the management of cancer, neurodegeneration, inflammation, and even the newly emerged pandemic, COVID-19 [9].

Structurally, there are four homologues of p38 MAPK including p38α, p38β, p38γ, and p38δ [3]. Among these, p38α is the best characterized and seems to be the most physiologically related protein involved in inflammatory responses [4,10]. According to the site of the ligand modulation, there are two different generations of p38α MAPK inhibitors, including type I and type II inhibitors, which modulate the activity of the enzyme at the ATP-binding and the allosteric site, respectively. However, targeting an ATP-binding site has limited the clinical use due to a high level of sequence and structural similarity among kinase enzymes [2], which could result in non- or low selective behavior and cause undesirable side effects and toxicities [11]. In order to overcome this issue, recent research has been focusing on utilizing a novel allosteric regulatory site, which is distinct from the ATP pocket at about 60° spatially, and there is no structural overlap between compounds bound to the allosteric site and ATP [12]. The conserved residues Asp-Phe-Gly (DFG) motif in the active site were conformationally altered, which is often known as DFG-out conformation and seems to be more stable in protein Tyr kinases [12]. To date, even though a number of clinical p38 MAPK inhibitors have emerged for inflammatory disease indications such as rheumatoid arthritis, there have been no approved agents [13,14] due to the lack of target modulation, adverse events, toxicities, and poor pharmacokinetics [4,14]. Some toxicities reported by clinical studies of well-known p38 MAPK inhibitors, BIRB796 (doramapimod), VX-745 (Vertex), and SCIO-469 (talmapimod) included hepatotoxic elevation of liver transaminases, skin rash, and so forth [15,16,17]. Accordingly, searching for novel compounds capable of impeding p38 MAPK has still been necessarily important to provide bottom-up preclinical information, guiding the development of therapeutic agents disrupting the MAPK signaling pathway.

Herein, by shedding light on the advancement of computational biology partly contributed to a preclinical stage of drug discovery and development, we aimed to search for novel agents capable of binding to p38α MAPK at the allosteric site by a drug repositioning approach. Bioinformatic databases and in silico methods including docking-based virtual screening, molecular dynamics (MD) simulations, and free energy calculations were employed to guide the discovery of hit compounds that may present a significant potential for further optimization. All the results obtained here provide some useful information and may outline the next steps governing experimental studies for drug discovery and development against p38α MAPK.

## 2. Materials and Methods

### 2.1. Preparation of the 3D Structure of P38α MAPK and Ligands

The three-dimensional structure of p38α MAPK in complex with a known inhibitor, BIRB796, was retrieved from the RCSB Protein Data Bank (PDB ID: 1KV2). The missing residues (170–184) of p38α MAPK were constructed by means of the homology model implemented in the SWISS-MODEL server [18]. The newly generated structure was consequently validated by plotting the Ramachandran diagram (Appendix A), using PROCHECK [19]. The protonation states of all ionizable amino acids were predicted based on their pKa value by using the PROPKA 3.0 web interface [20], and were then set into the modeled complex structure before performing molecular docking and MD simulations.

Partial atomic charges of BIRB796 were calculated for their geometry and then assigned for the electrostatic potential (ESP) charges via the Gaussian 09W program (G09) using the Hartree–Fock method and 6-31G(d) level of theory [21]. Its structure was then assigned for atom type, and we generated its topology file using the Antechamber program [22]. Converting from the ‘mol2’ file into a ‘pdbqt’ format was achieved by AutoDockTools (ADT). For the virtual screening studies, all focused compounds were obtained from the 3210 FDA-approved drugs available in the ZINC database (http://zinc.docking.org, accessed on 12 November 2019). These ligands were also subsequently converted from the ‘mol2’ format into a ‘pdbqt’ format using ADT.

### 2.2. Molecular Docking and Visual Inspection

Docking calculations were carried out on a Linux operating system using AutoDock VinaXB, which provides a new empirical halogen bond scoring function [23]. Three docking parameters, including exhaustiveness, num_modes, and energy_range, were set to 20, 50, and 5 kcal/mol, respectively. For system validation, the crystallized ligand was redocked into the same binding site (Appendix A), and the verified grid box was then employed for all ligands in virtual screening. Predicted binding affinity (E_binding_, kcal/mol) of the most likely occurring conformation was a parameter used to rank the studied compounds, and the structure coordinate was employed to be the initial structure for the MD run.

To reduce the chance of false-positive scoring, a visual inspection of the intermolecular interactions between each ligand and amino-acid residues lining in the focused allosteric site was carried out by specifically examining (i) hydrogen bonding with E71, M109, and D168 as well as (ii) hydrophobic interactions with V38, A51, K53, R67, L75, I84, L104, L108, A157, L167, and F169, which were derived from the binding mode observed in the inhibitor prototype, BIRB796. For this purpose, compounds sharing features of intermolecular interactions with BIRB796 greater than five interactions were then selected.

### 2.3. Molecular Dynamics (MD) Simulations

The protein-ligand complex coordinates of the two screened drugs and the inhibitor prototype from molecular docking were dynamically modeled under the periodic boundary condition with the isothermal-isobaric (NPT) scheme [24,25,26,27,28]. The AMBER ff14SB and generalized AMBER force field version 2 (GAFF2) [29] were selected for a force field governing bonded and nonbonded interaction parameters. Electrostatic interactions were treated by the particle mesh Ewald summation method [30] with a cutoff distance for nonbonded interactions of 10 Å. The SHAKE algorithm [31] was retrieved to constrain hydrogen atoms. The temperature was controlled by the Langevin thermostat [32] and set to 310 K by increasing from 10 to 310 K. Controlling the pressure was achieved by the Berendsen barostat [33] with a relaxation time of 1 ps. Moreover, the TIP3P water model [34] was used to solvate the system with minimum padding of 10.0 Å between the protein surface and the solvation box edge. The overall charge of the molecular system was neutralized by randomly adding either sodium or chloride ions. Minimization of the added hydrogen atoms and water molecules was carried out using 500 steps of steepest descent (SD) followed by 1500 steps of conjugated gradient (CG) methods before running the MD simulations with constrained solvent molecules. The whole complex was then fully minimized using the same procedure. For MD production, all systems were set to 100 ns (2-fs increment). The root-mean-square displacement (RMSD), the numbers of hydrogen bond (H-bond), and the contact atoms were calculated through the cpptraj module whilst the per-residue decomposition energy (
${\Delta G}_{\mathrm{binding}}^{\mathrm{residue}}$
) was estimated by MM/PBSA.py implemented in AMBER16.

### 2.4. End-Point Binding Energy Calculations

To observe the ligand-binding affinity, the end-point binding free energy (∆G_bind_) of each system was predicted by the solvated interaction energy (SIE) approach [35]. ∆G_bind_ can be estimated by the summation of the van der Waals (E_vdW_), electrostatic (E_ele_), reaction field (G_RF_), cavity (γΔSA(ρ)), and a constant (C) value. The mathematical equation can be expressed as follow:∆G_bind_ (ρ, D_in_, α, γ, C) = α[E_vdW_ + E_ele_(D_in_) + ΔG_RF_(ρ,D_in_) + γΔSA(ρ)] + C
where D_in_ denotes the solute dielectric value. E_vdW_ and E_ele_ represent intermolecular van der Waals and Coulombic interaction energies in the bound state, respectively. ΔG_RF_ is the alteration of the reaction field energy between the bound and free states, ΔG_cavity_ (γΔSA) denotes the change in the non-electrostatic solvation free energy between the bound and free forms, and C is the constant value. The coefficients were set as α = 0.105, γ = 0.013, and C = −2.89.

### 2.5. QM-Based ONIOM Binding Energy Calculations

A quantum mechanics (QM)-based Our Own N-layered Integrated Molecular Orbital and Molecular mechanics (ONIOM) [36,37] was carried out to additionally observe the binding strength between BIRB796 and the screened drug candidate(s). Before calculating the binding energy, the constructed complexes were optimized using the Hartree−Fock method and a mechanical parameter (HF/6-31G(d):UFF). After optimization, two-layered ONIOM calculations (B3LYP/6-31G(d,):PM6) were applied to determine and compare the binding energies of the three systems. The residues lining within the 5 Å from the ligand, which include Y35, V38, A51, V52, K53, L55, R67, T68, R70, E71, L74, L75, I84, L86, L104, V105, T106, H107, M109, H148, R149, L167, D168, F169, G107, and L171, were selected to represent an allosteric site of p38α MAPK and separated into a low-level layer, while each screened drug was set to a high-level layer. Then, the selected amino acid residues and the ligand were again simulated individually with the B3LYP/6-31G(d) basis set and the PM6 method, respectively. The polarizable continuum model (PCM) was applied to observe the effect of water solvent on the binding energy. All calculations were performed by using the GAUSSIAN16 software package [38], and the binding energy was estimated using the equation below [39].

$$E_{\mathrm{binding}}^{\mathrm{solvation}}=E_{\mathrm{complex}}^{\mathrm{PCM}}\;-\;E_{\mathrm{residues}}^{\mathrm{PCM}}-E_{\mathrm{ligand}}^{\mathrm{PCM}}$$

where 
$E_{\mathrm{binding}}^{\mathrm{solvation}}$
 is the binding energy of the drug-receptor in the solvation system, 
$E_{\mathrm{complex}}^{\mathrm{PCM}}$
 is the extrapolated ONIOM energy of the complex, 
$E_{\mathrm{residues}}^{\mathrm{PCM}}$
 is the potential energy of residues lining within the 5 Å from the ligand, and 
$E_{\mathrm{ligand}}^{\mathrm{PCM}}$
 is the potential energy of the studied ligand.

## 3. Results and Discussion

### 3.1. Docking-Based Screening and Visual Inspection

Finding the existing drugs that can offer inhibition towards novel targets is a great challenge. To this end, 3210 compounds retrieved from the FDA-approved drugs available in the ZINC database were docked into the allosteric site of the p38α MAPK where its 3D-structure and the inhibitor binding site are illustrated in Figure 1. The compounds were selected and ranked according to their binding affinity (E_binding_) predicted by the scoring function of the Autodock XB software package. By considering their binding affinity, it was found that, among the 3210 compounds, only ZINC27990463 exhibited higher binding affinity (E_binding_ = −12.10 kcal/mol) when compared to the ligand reference, BIRB796 (E_binding_ = −11.9 kcal/mol). However, to reduce false-negative selection, compounds exhibiting E_binding_ lower than −10.00 kcal/mol were also clustered, which were totally filtered into 22 compounds. Note that the in silico filtering scheme and the plot of binding affinity of the selected first-round screened compounds were illustrated in Figure 2 and Figure 3, respectively. All these first-round screened compounds were then inspected for intermolecular interactions inside the cleft of the allosteric site, which is commonly known as “visual inspection”. This method has been widely used in the decision-making step for a great number of drug discovery campaigns [40].

It is obviously known that noncovalent interactions are essentially responsible for ligand binding. Compounds showing sufficient interactions in both qualitative and quantitative manners tend to exhibit greater binding capability and could form a more stable complex. Thus, an inspection of intermolecular interactions between each ligand and amino-acid residues lining in the focused allosteric site was visually carried out by specifically examining (i) hydrogen bonding with E71, M109, and D168, as well as (ii) hydrophobic interactions with V38, A51, K53, R67, L75, I84, L104, L108, A157, L167, and F169, which were derived from the binding mode observed in the inhibitor prototype, BIRB796. For this purpose, compounds sharing features of intermolecular interactions with BIRB796 greater than five interactions were then selected, for which the detailed information of all 22 compounds is listed in Figure 4. Thus, we could obtain 10 promising compounds (Figure 2A), which are hereinafter referred to as “hit compounds”. These 10 hit compounds were subsequently subjected to second-round screening by MD simulations, and the MD output was used to compute SIE-based end-point free energy calculations.

### 3.2. Dynamic-Based Screening and End-Point Binding Free Energy Calculations

To observe and screen the hit compounds’ binding capability in a near-physiological condition and dynamic system, the constructed protein-ligand complexes were performed to run MD simulations for 100 ns. The trajectories in the last ten nanoseconds (90–100 ns) was considered to have reached the equilibrated state (supported by the plot of root-mean-square displacement (RMSD) for the backbone amino acids within 5 Å from the ligand as shown in Appendix A) were used to calculate the binding free energy (∆G_bind_). This parameter was used to indicate the protein-ligand binding affinity and to employ a dynamics-based screening tool for ranking the hit compounds in the aftermath of rigid docking.

As listed in Table 1, the ∆G_bind_ values of all hit compounds were in the range of −11.4 to −7.2 kcal/mol, whilst the ∆G_bind_ of BIRB796 is −11.95 kcal/mol. Importantly, the predicted ∆G_bind_ of BIRB796 (−11.95 ± 0.04 kcal/mol) is close to the experimental-derived ∆G_bind_ value (−10.98 kcal/mol [41]), showing the verification of the predictive method and the reliability of the results obtained. For screening purposes, only two hit compounds, lomitapide (∆G_bind_ = −11.39 ± 0.05 kcal/mol) and nilotinib (∆G_bind_ = −11.27 ± 0.03 kcal/mol) displaying a similar level of binding strength to the BIRB796, were selected for further investigation and the chemical structures of these three drug candidates are illustrated in Figure 2B. For nilotinib, it was previously reported that it could be a new off-target to p38 MAPK in the myoblast cell line [42], which could support our theoretical findings. In particular, the calculated energy terms shown in Table 1 could imply the influence of specific types of noncovalent interactions responsible for drug recognition. In this case, we found that all three drugs possessed a considerably higher contribution of van der Waals interaction energies than other types of interaction energies, agreeing well with the previous study that suggested the hydrophobicity of the binding pocket [43]. Additionally, the higher contribution of vdW interaction energies might imply that the screened drugs could preferentially target the hydrophobic regions within the focused binding site, which was similarly observed in the previously reported potent inhibitors [11]. For the solvation effect, the polar solvation energies expressed as the ΔG_RF_ were in the range of 10.52 to 20.55 kcal/mol.
Lomitapide and nilotinib showed a slightly higher ΔG_RF_ (17.01 ± 0.24 and 17.15 ± 0.17, respectively) than BIRB796 (15.60 ± 0.20), implying the relatively minute higher polar solvation in the lomitapide and nilotinib complex system. For ΔG_cavity_, it was found that the nonpolar solvation energies were in the range of −7.13 to −14.43 kcal/mol. Among candidates, lomitapide possessed the highest contribution of ΔG_cavity_ (−14.43 ± 0.04), showing that the drug could be well-buried into the cleft of the binding site while nilotinib and BIRB796 demonstrated a slight reduction in the nonpolar solvation effect (−12.81 ± 0.04 and −13.63 ± 0.04, respectively). By including the solvation free energy, the vdW term (ΔE_vdW_ + 
$∆G_{\mathrm{sol}}^{\mathrm{nonpolar}}$
) was the main contribution to the total binding free energies of both drug candidates as well as BIRB796 whilst the electrostatic term (ΔE_ele_ + 
$∆G_{\mathrm{sol}}^{\mathrm{polar}}$
) became much less favorable to the binding (Appendix A).

### 3.3. Contact Atoms and Numbers of Hydrogen Bond Formation

Identifying the number of atoms surrounding a ligand is one of the crucial parameters implying the ability of the drug recognition within the focused allosteric target. Herein, noncovalent contacts of any atoms within the 5.0 Å from the ligand were computed, and we found that the number of surrounding atoms averaged in the last 10 nanoseconds of each focused complex was in the order of lomitapide (429 ± 17 atoms) > BIRB796 (424 ± 6 atoms) > nilotinib (396 ± 19 atoms) as illustrated in Figure 5. The number of surrounding atoms of the lomitapide complex was slightly higher than that of the BIRB796 and nilotinib complexes, suggesting that the binding pocket residues are close-packed during complexation.

Furthermore, the quantity of hydrogen bonds (H-bond), which was considered one of the strong interactions responsible for drug-receptor binding, was analyzed during 90–100 ns with three independent replicates. As shown in Figure 5, the numbers of averaged H-bond interactions in BIRB796 were approximately 4 bonds while lomitapide and nilotinib were in a vicinity of 2 and 3 bonds, respectively. For drug binding, this indicated that both BIRB796 and nilotinib could form more H-bonds when compared to lomitapide. This is likely to occur since the intrinsic structural characteristic of lomitapide consists of gradual lower numbers of hydrogen bond donors and acceptors when compared to BIRB796 and nilotinib, (total numbers of H-bond donors and acceptors of BIRB796, nilotinib, and lomitapide are 7, 8, and 5, respectively (analyzed by PharmaGist web interface [44] as listed in Appendix A).

In addition, the intermolecular H-bond interactions were observed in terms of the percentage of occupations (Figure 5B), which indicated how often the transient H-bonds could occur during the whole simulated time. As expected, a few strong hydrogen bonds could be seen in all focused drugs and even in BIRB796 since their inhibitory actions were mainly driven by hydrophobic interactions (Table 1). Obviously, all three compounds were found to have very strong H-bonds with D168 (99.7%, 92.7%, and 97.2% occupations for BIRB796, lomitapide, and nilotinib, respectively). Additionally, the BIRB796 and nilotinib could form an additional very strong H-bond with E71 (2 H-bonds for BIRB796 and 1 H-bond for nilotinib). The slight loss of this interaction in nilotinib might cause a slight reduction in binding affinity when compared to BIRB796 (Table 1). Nonetheless, we could not observe the H-bond with E71 for lomitapide binding since it lacks H-bond donors at that oriented position. Hence, we hypothesized that adding functional groups containing H-bond donors (e.g., -NH_2_) onto the carbon atom in the piperidine ring within its structure might allow it to have more additional H-bond interactions with E71. Hence, we ran MD simulations of the modified structure of lomitapide and subjected the MD output to analyze the binding energy at the last 10 ns by using the end-point SIE-based method (as the same protocol used previously with other compounds). As expected, the H-bond occupation with E71 could be formed at 82.45% during the whole simulated time. Moreover, the binding energy was decreased from −11.39 ± 0.05 kcal/mol to −12.15 ± 0.06 kcal/mol (better binding affinity, Appendix A), and slightly lower than BIRB796 (−11.95 ± 0.04 kcal/mol). This finding suggested that modification of a lomitapide’s structure by permitting it to interact with E71 could improve its binding affinity, which encouraged us to investigate further.

### 3.4. Key Binding Residues

To elucidate the key binding amino acids responsible for the drug recognition within the allosteric pocket of p38α MAPK, the decomposition of free energy (
${\Delta G}_{\mathrm{residue}}^{\mathrm{bind}}$
) based on the MM/GBSA method was computed. The negative and positive 
${\Delta G}_{\mathrm{residue}}^{\mathrm{bind}}$
 values indicate the ligand stabilization and destabilization, respectively. The contribution of each amino acid of the known inhibitor and two focused complexes is shown in Figure 6. Note that among residues 5–352 of p38α MAPK, only residues 5–250 are shown.

For the BIRB796, it was obviously seen that E71 and D168 played a pivotal role in stabilizing the protein–ligand complex as its large 
${\Delta G}_{\mathrm{residue}}^{\mathrm{bind}}$
 value was observed (approximately −6 and −4 kcal/mol, respectively). In addition, four hydrophobic residues (L75, I84, L108, and L107) and one polar uncharged amino acid (T106) were found to be involved in a process of complex formation. This key-binding elucidation agreed well with the previous reports of BIRB796’s binding mode analysis [11,41]. Apart from a reference ligand, the amino acids largely contributing to the lomitapide binding (
${\Delta G}_{\mathrm{residue}}^{\mathrm{bind}}$
 < −1.5 kcal/mol) include L74, L75, I84, T106, L167, L171, and H174. Almost all were hydrophobic residues (except H174, a polar positively charged residue), suggesting the ensembles of hydrophobic interactions were dominant towards the binding (supported by per-residue vdW interaction energy as illustrated in Appendix A). Among these, the four residues L75, I84, T106, and L167 shared binding features in common with BIRB796. Interestingly, unlike BIRB796, lomitapide could bind to L171 and H174. Having interaction with the amino acids in the region of 170–199 attracted considerable attention, as similarly observed in a new series of benzooxadiazole-based p38 inhibitors which was granted a patent in 2014–2015 (Allinky Biopharma. Co., Madrid, Spain) [11]. In the case of nilotinib, it was found that key amino acids contributing to its binding were mostly the same residues responsible for BIRB796 binding (E71, I84, L167, and D168) since it belongs to the same type of inhibitor (kinase inhibitor). Among these, E71 and D168 were essentially responsible for stabilizing the complex via H-bond while the others relied on hydrophobic interactions (Appendix A). Two additional residues, L74 and M109 were also observed. We noted that these results are correlated well with the calculated SIE-based ΔG_bind_ and each energy component as listed in Table 1.

### 3.5. QM-Based ONIOM Binding Energy

The analysis of the ONIOM binding energies was employed to additionally observe the binding ability of the two screened drugs within the focused allosteric site of p38α MAPK. The calculations were based on the QM method, which could provide a more reliable prediction when compared to the end-point estimation [45]. As shown in Table 2, the calculated binding energy (
$E_{\mathrm{binding}}^{\mathrm{solvation}}$
) values ranged from approximately −41.0 to −48.5 kcal/mol. The 
$E_{\mathrm{binding}}^{\mathrm{solvation}}$
 values displayed a similar trend to the SIE-based prediction in which the binding affinity of BIRB7996 was slightly higher than that of lomitapide and nilotinib. Even though the 
$E_{\mathrm{binding}}^{\mathrm{solvation}}$
 of the two screened drugs showed as slightly lower, their predicted binding strength was still high and comparable to the reference inhibitor, BIRB796. Accordingly, we believe that lomitapide and nilotinib could be able to inhibit the phosphorylation of p38α MAPK, and the ONIOM-based method theoretically confirmed their inhibitory capability towards p38α MAPK at the allosteric site. It is worth noting that the prediction trend of binding affinity by ONIOM energy calculations was in good agreement with the SIE-based end-point method.

## 4. Conclusions

Since there have been no drugs approved as therapeutic agents for p38α MAPK, our research is considered one of the collective efforts to search for effective drugs targeting this target at the allosteric site by a drug repurposing approach. Verified docking- and dynamic-based screening revealed that lomitapide and nilotinib could alternatively impede the p38α MAPK’s function with a great binding affinity and characteristics. The binding affinity estimated by both end-point and QM-based ONIOM methods revealed a comparable level to the inhibitor prototype (BIRB796), supported by the calculated numbers of atoms surrounded within the 5.0 Å from the ligand. Specifically, vdW interaction energies were the main force driving the complex formation. Moreover, all drugs could form a few H-bonds with the amino acids lining in the allosteric site, which could rank in the order of BIRB796 ≈ nilotinib > lomitapide. The two residues (E71 and D168) played a pivotal role in forming very strong H-bonds with the focused drugs. More importantly, we proposed that modifying a lomitapide’s structure by allowing it to interact with E71 via H-bonds could improve its binding affinity. Altogether, our in silico study not only presented the potential inhibitors, but also provided useful information at the atomic level to shed light on rationally designing more potent inhibitors disrupting the MAPK signaling pathway. However, experiments determining the biological activities of these elucidated compounds including enzyme- and cell-based assays should be further carried out.

## Figures and Tables

**Figure 1 pharmaceutics-14-01461-f001:**
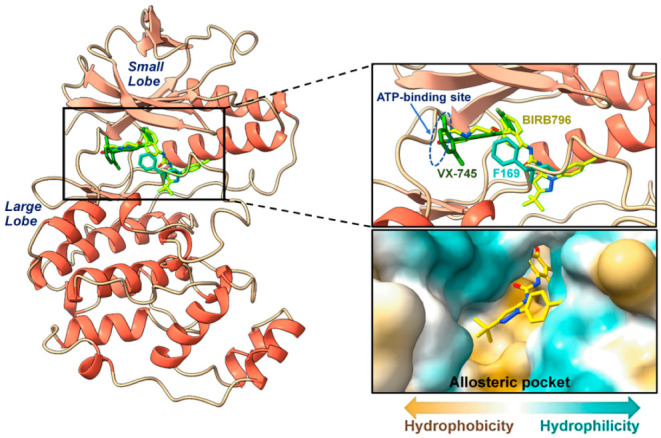
The ribbon representation of the 3D structure of p38α MAPK (PDB ID: 1KV2). The close-up regions illustrate two common inhibitors including VX-745 (green) and BIBR796 (yellow), indicating the ATP-binding site and allosteric pocket, which is distinct from each other at about 60° spatially. An orientation of F169 exhibiting the unique DFG-out conformation is also shown. Additionally, the hydrophobic nature (obtained via UCSF ChimeraX 1.4, Resource for Biocomputing, Visualization, and Informatics (RBVI), San Francisco, CA, USA.) within the focused allosteric cleft is depicted in a close-up view.

**Figure 2 pharmaceutics-14-01461-f002:**
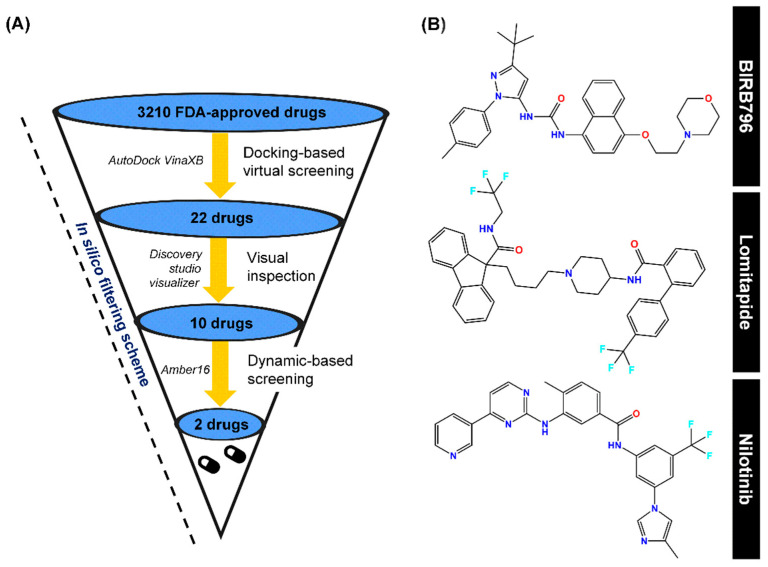
(**A**) In silico filtering scheme, which includes first-round docking-based screening, visual inspection, and SIE-based dynamic screening as well as the program used during each step. (**B**) Chemical structures of a well-known p38α MAPK allosteric inhibitor (BIRB796), Lomitapide, and Nilotinib, obtained via this computational platform.

**Figure 3 pharmaceutics-14-01461-f003:**
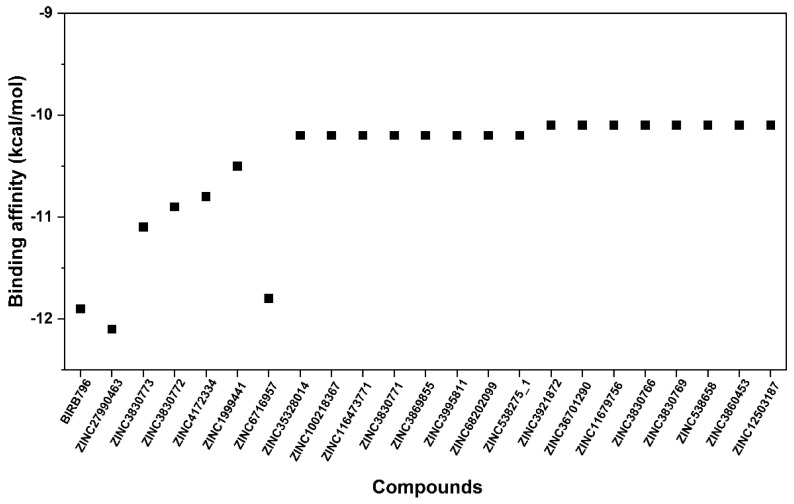
Binding affinity in kcal/mol of selected first-round screened compounds that were successfully docked into the focused allosteric site of p38α MAPK compared to BIRB796. Note that the prediction was based upon the scoring function implemented in the Autodock VinaXB, Sirimulla Research Group at the University of Texas at El Paso, TX, USA.

**Figure 4 pharmaceutics-14-01461-f004:**
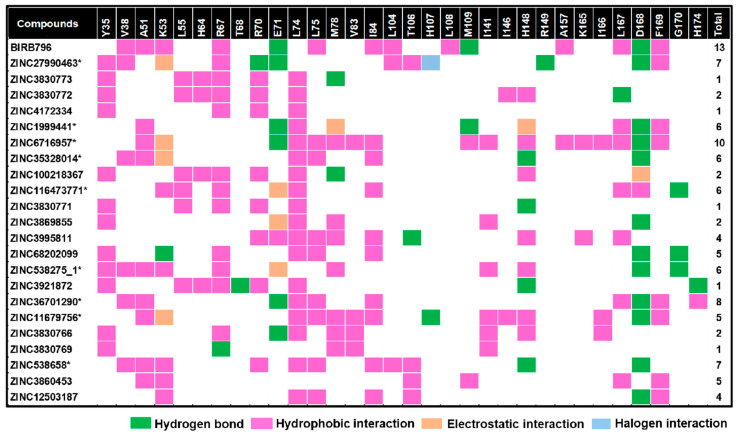
Map of intermolecular interactions of BIRB796 and all 22 screened compounds as well as total features sharing interactions with BIRB796. Each type of noncovalent interaction was also illustrated in different colors. It is worth noting that these occurred interactions were based upon the best docked conformation and visualized by Accelrys Discovery Studio 2.5. * The compounds selected to run MD simulations.

**Figure 5 pharmaceutics-14-01461-f005:**
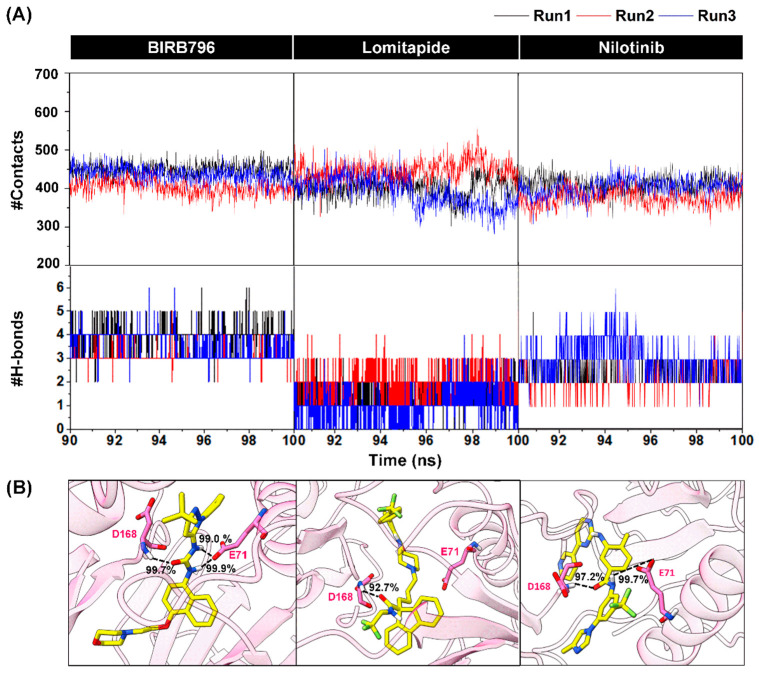
(**A**) Numbers of surrounding atoms counted within the 5.0 Å from the ligand and number of H-bonds within p38α MAPK-BIRB796 complex and two focused drugs at the last 10 nanoseconds (90–100 ns). The results were shown in three independent runs. (**B**) Percentage of H-bond occurrence during a complex formation of two screened drugs and the BIRB796 using two criteria as follows: (1) the distance between the hydrogen bond donor (HD) and hydrogen acceptor (HA) of ≤3.5 Å (2) the angle ≥120°.

**Figure 6 pharmaceutics-14-01461-f006:**
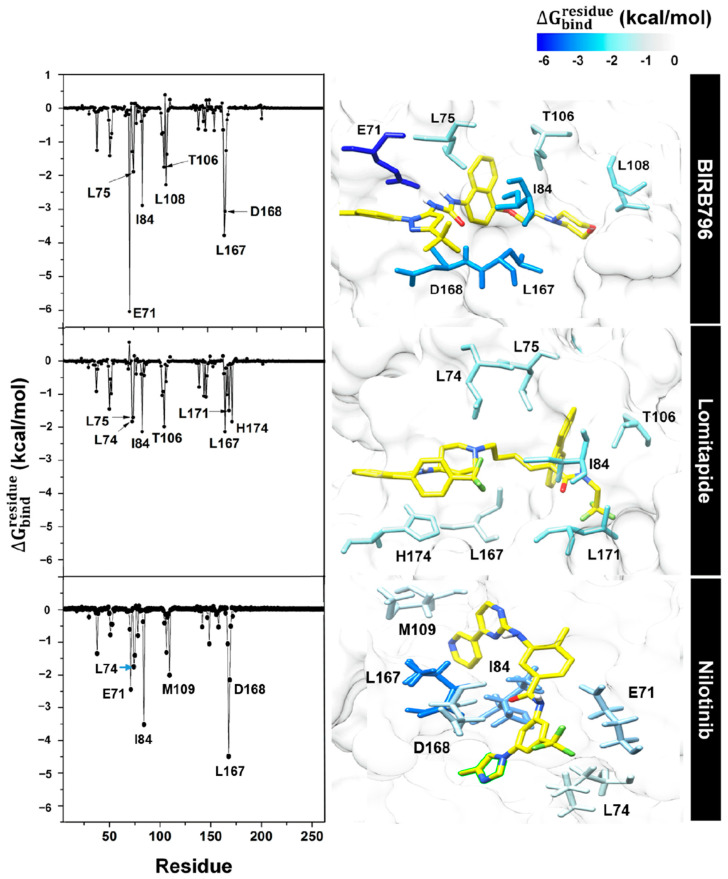
Per-residue free energy decomposition of amino acids involved in ligand binding where the highest to lowest 
${\Delta G}_{\mathrm{residue}}^{\mathrm{bind}}$
 contribution (more negative value) was shaded from dark blue to white.

**Table 1 pharmaceutics-14-01461-t001:** ΔG_bind_ values (kcal/mol) of the candidate compounds as well as BIRB796 in complex with p38α MAPK calculated by the SIE-based end-point method using α, γ, and constant coefficients of 0.10, 0.01, and −2.89, respectively.

Drugs (ZINC ID)	Energy Components (kcal/mol)				
	E_VdW_	E_coul_	ΔG_RF_	ΔG_cavity_	ΔG_bind_
BIRB796	*experiment*				−10.98 *
	−78.53 ± 0.29	−9.93 ± 0.15	15.60 ± 0.20	−13.63 ± 0.04	−11.95 ± 0.04
Lomitapide (ZINC27990463)	−77.77 ± 0.39	−5.95 ± 0.18	17.01 ± 0.24	−14.43 ± 0.04	−11.39 ± 0.05
Nebivolol (ZINC1999441)	−49.57 ± 0.29	−12.66 ± 0.26	12.78 ± 0.21	−10.19 ± 0.03	−9.14 ± 0.03
Nilotinib (ZINC6716957)	−71.93 ± 0.26	−12.36 ± 0.19	17.15 ± 0.17	−12.81 ± 0.04	−11.27 ± 0.03
Ibrutinib (ZINC35328014)	−61.87 ± 0.29	−4.07 ± 0.17	13.21 ± 0.23	−10.81 ± 0.04	−9.55 ± 0.04
Atovaquone (ZINC116473771)	−42.15 ± 0.27	−2.43 ± 0.39	10.94 ± 0.23	−7.90 ± 0.05	−7.24 ± 0.03
Dicumarol (ZINC3869855)	−39.68 ± 0.26	−13.87 ± 0.40	20.55 ± 0.32	−7.13 ± 0.03	−7.09 ± 0.03
Raloxifene (ZINC538275)	−51.19 ± 0.44	−12.45 ± 0.50	18.50 ± 0.25	−9.90 ± 0.05	−8.66 ± 0.07
Ponatinib (ZINC36701290)	−61.66 ± 0.25	−4.92 ± 0.14	16.99 ± 0.24	−12.08 ± 0.05	−9.35 ± 0.03
Eltrombopag (ZINC11679756)	−59.44 ± 0.33	−5.39 ± 0.17	10.52 ± 0.17	−10.97 ± 0.03	−9.73 ± 0.04
Samsca (ZINC538658)	−51.69 ± 0.26	−16.63 ± 0.21	19.86 ± 0.18	−10.38 ± 0.04	−9.05 ± 0.03

* The experimental binding free energy was derived from the IC_50_ of 0.018 μM [41] and was calculated by the equation 
${\Delta G}_{\mathrm{bind}\;}={\;\mathrm{RTlnIC}}_{50}$
.

**Table 2 pharmaceutics-14-01461-t002:** Calculated binding energy (
$E_{\mathrm{binding}}^{\mathrm{solvation}}$
) in kcal/mol of two screened drugs and BIRB796 by means of ONIOM at B3LYP/6-31G(d):PM6 level of theory.

**Drugs**	Energy Terms			
	$E_{\mathrm{complex}}^{\mathrm{PCM}}$ (a.u.)	$E_{\mathrm{residues}}^{\mathrm{PCM}}$ (a.u.)	$E_{\mathrm{ligand}}^{\mathrm{PCM}}$ (a.u.)	$E_{\mathrm{binding}}^{\mathrm{solvation}}$ (kcal/mol)
BIRB796	−1706.396	−3.391	−1702.927	−48.536
Lomitapide	−2425.571	−3.303	−2422.203	−41.159
Nilotinib	−1841.536	−3.240	−1838.230	−41.048

## Data Availability

Not applicable.

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
