# Peer review of "Integration of In Silico Strategies for Drug Repositioning towards P38α Mitogen-Activated Protein Kinase (MAPK) at the Allosteric Site"

_pharmaceutics, 2022, doi:10.3390/pharmaceutics14071461_

Round 1
Reviewer 1 Report
This manuscript details the use of molecular docking and all-atom molecular dynamics (MD) simulations to perform an in silico screening of a library of small-molecule FDA-approved drugs to propose possible inhibitors of p38alpha MAPK, a key protein involved in numerous physiological pathways including cell proliferation.
The work follows well-established protocols involving automated docking, atomistic simulations and approximate free energy calculations based on an solvated interaction energy (SIE) approach. The work is competently carried out, well written and presented, and I recommend publication in its present form.
My only comment is that all of the figures in the PDF file I reviewed are of rather low resolution, and the authors should endeavour to enhance the graphical quality prior to re-submission.
Author Response
Point 1: My only comment is that all of the figures in the PDF file I reviewed are of rather low resolution, and the authors should endeavor to enhance the graphical quality prior to re-submission.
Response 1: Thank you very much for giving positive opinions about our manuscript. For the graphical quality, we enhanced the graphical resolution from 300 dpi to 600 dpi for re-submission.
Reviewer 2 Report
The study reported by Suriya, et al. utilized molecular docking to screen the potential p38a MAPK inhibitors from 3210 FDA-approved drugs and MD simulations and MM/PBSA binding free energy analysis for refinement. Finally, two screened drugs were determined as potential compounds for developing therapeutic agents targeting p38a MAPK. Although the research topic is interesting and crucial, the methods and analyses used in this study are quite common, nothing special, and without further enzyme or cellular experiments to validate the in silico screening results. It is highly possible to obtain the negative results for the two screened compounds. To my knowledge, the binding free energy calculations are still not absolutely precise prediction in drug screening and discovery field, there are still some biased cases. Detailed comments for this manuscript are shown in the following,
1. line 106: please give the full name of ESP charges
2. Figure 1: for the hydrophobic distribution, please describe how to obtain the map by which software.
3. line 142: ....whist the per-residue......, please check any typo for "whist"
4. line 176: the author mentioned "visual inspection", please also describe in detail in the "methods" section.
5. The VDWs interaction is not equaled to hydrophobic interactions, but the authors seem to emphasize the VDW contribution as to hydrophobic interactions. How about the slovation contribution?
6. The authors finally determine the two potential drugs by MM/PBSA analysis, why not use FEP (free energy perturbation) calculation for the two compounds?
Reviewer 3 Report
The manuscript entitled (Integration of In Silico Strategies for Drug Repositioning towards P38α Mitogen-Activated Protein Kinase (MAPK) at the Allosteric Site) by Suryya et al. presented a docking model and 100-ns dynamic-based screening. Some compounds were found to theoretically inhibit phosphorylation of p38α MAPK at the allosteric site. The molecular dynamics simulations and free energy calculations showed that the binding affinity of the two screened drugs exhibited a similar level as the known p38α MAPK inhibitor (BIRB796), suggesting a high potential of being a novel p38α MAPK inhibitor. However, all these findings remain theoretical conclusions not based on wet experiments. I suggest authors test their compounds on the molecular level of the p38-alpha enzyme and resubmit the manuscript. Another major issue in this work is that the resolution of the figures presented in the paper is too low. Higher-resolution figures are mandatory.
Reviewer 4 Report
The manuscript by Suriya et al. presents an in silico study directed to the search for inhibitors of p38 MAPK among known drugs. In this drug repurposing investigation, the authors applied methodology based on docking and molecular dynamics simulations. The obtained data were thoroughly analyzed in terms of binding energy, per-residue and per-atom partial free energies. Two compounds were eventually selected which are perspective as therapeutic agents targeting p38α MAPK. The results are important and deserve publication.
Specific comments:
Lines 25-26: Bad phrase: "...interesting charged residues E71 and 25 D168 were of particular interest...".
Line 151: The meaning of "constant" in the equation is not clear. Is this the same as "C" in the text below the equation? Please clarify.
Line 200: The term "rigid-docked conformation" is ambiguous. Obviously, the authors imply that the protein was considered rigid on docking, while the ligands were conformationally flexible. Please replace this phrase with a more suitable one.
Line 207: The authors should clearly indicate where the 5-A sphere was centered (at the geometric center of the ligand?). The same refers to lines 236, 269, 317, and to the caption of Figure S3. Maybe, the authors mean "within 5 A from the ligand" (not sphere)?
Line 231: The equation is quite strange. According to this equation, the lower is IC50, the higher (or less negative) is DeltaG. Perhaps, it is necessary to remove the minus sign, as the equilibrium constant of the complex formation actually is inversely proportional to IC50.
Line 266: Please replace "C atom" with "carbon atom".
Line 277: The phrase "decomposition free energy" is inappropriate here. Conventionally, it denotes free energy of bond breaking on a molecule destroying. The authors mean decomposition of deltaG value into individual terms. Hence, "free energy decomposition" or "decomposition of free energy" would be more correct. Please check throughout the manuscript and the supplementary file.
Line 321: Please replace "proposed modifying" with "proposed that modifying".
Quality of figures is low. Please enhance resolution.
Figures 1 and 2 are not mentioned in the text of the manuscript.
Caption of Figure S3: For each color, it is desirable to indicate initial velocities of the MD runs.
Summarizing, I recommend acceptance of the manuscript for publication after minor revision.
Author Response
Dear reviewer #4
Please see the attachment

Round 2
Reviewer 2 Report
Although the authors added more explanations in the revised manuscript, the methods used in the study are quite general, and due to the lack of cellular experiments to confirm the modeling results, it would be better if the authors could perform another binding free energy calculation or decoy docking for the two compounds to double check the modeling results. It is quite possible to obtain the biased data for MM/PBSA calculation.
By the way, two compounds were determined to be similar to the inhibitor BIRB796 with similar calculated binding energy. However, the binding energies were still a bit higher that BIRB796, the modification of the the two lead compounds should be crucial, and the authors should have more discussion on this or run the binding energy for the modified structures.
Author Response
Dear reviewer,
Please see the attachment
With best regards,
On behalf of the authors,
Thanyada Rungrotmongkol

Reviewer 3 Report
If the special issue where authors submitted the manuscript accept only in silico investigations, then no problem to accept this paper in this special issue. However, In this case, I suggest authors to find any related published in vitro results that could support their theoritical findings and cite this work.
Author Response
Response to reviewer 3
Point 1: In this case, I suggest authors find any related published in vitro results that could support their theoretical findings and cite this work.
Response 1: Thank you very much for your valuable suggestion. We have cited the available papers involving in vitro assay of BIRB796 (0.018 mM of IC50, citation 41). Even though we didn’t determine the IC50 value to compare with this paper, we could convert it to the experimental ΔGbind value by Cheng-Prusoff equation [1]; ΔGbind = RTlnIC50. In comparison to our calculated ΔGbind value, we discussed in section 3.2 that the predicted ∆Gbind of the BIRB796 (-11.95±0.04 kcal/mol) is close to the experimental-derived ∆Gbind value (-10.98 kcal/mol), showing the verification of the predictive method used in this study. Moreover, one experimental study [2] reported that nilotinib could inhibit p38 phosphorylation in the myoblast cell line and the authors concluded that it could be a new off-target to p38 MAPK. This is one of the experimental shreds of evidence that supports our theoretical findings, which was cited in the revised manuscript [citation 42]. In addition, in citation 11, we discussed the common features of intermolecular interactions we found in screened compounds with already-proven potent p38 inhibitors in a series of benzooxadiazole; this could be supporting evidence to our finding as well. However, we still realized that experimental determination of the biological activities of these elucidated compounds should be further carried out. So, we left this sentence in the conclusion part.
Reference:
[1] Cheng, H.C., The power issue: Determination of KB or Ki from IC50: A closer look at the Cheng–Prusoff equation, the Schild plot and related power equations. Journal of pharmacological and toxicological methods, 2001, 46(2), p. 61-71.
[2] Contreras, O., M. Villarreal, and E. Brandan, Nilotinib impairs skeletal myogenesis by increasing myoblast proliferation. Skeletal Muscle, 2018. 8(1): p. 5.
Reviewer 4 Report
Thank you very much. The manuscript can be accepted for publication in the revised form.
Author Response
Dear reviewer,
Thank you very much for accepting our manuscript to be published in Pharmaceutics.
With best regards,
On behalf of the authors,
Thanyada Rungrotmongkol
Round 3
Reviewer 2 Report
The authors have answered my concerns for the manuscript, and run another quantum mechanical-based binding free energy calculation. The manuscript can be accepted for publication.
Reviewer 3 Report
My concerns were addressed